# Indole-3-Butyric Acid, a Natural Auxin, Protects against Fenton Reaction-Induced Oxidative Damage in Porcine Thyroid

**DOI:** 10.3390/nu16173010

**Published:** 2024-09-06

**Authors:** Anna K. Skoczyńska, Aleksandra K. Gładysz, Jan Stępniak, Małgorzata Karbownik-Lewińska

**Affiliations:** 1Department of Endocrinology and Metabolic Diseases, Medical University of Lodz, 281/289 Rzgowska St., 93-338 Lodz, Poland; anna.skoczynska@umed.lodz.pl (A.K.S.); aleksandra.gladysz@umed.lodz.pl (A.K.G.); jan.stepniak@umed.lodz.pl (J.S.); 2Polish Mother’s Memorial Hospital—Research Institute, 281/289 Rzgowska St., 93-338 Lodz, Poland

**Keywords:** indole-3-butyric acid, Fenton reaction, lipid peroxidation, antioxidant, porcine thyroid, reactive oxygen species, iron

## Abstract

We present results on the potential protective antioxidant properties of indole-3-butyric acid. Indole-3-butyric acid is an indole derivative defined as an auxin and widely known as a plant growth regulator. It naturally occurs in *Arabidopsis thaliana*, which is applied as a model plant in genetic studies. Oxidative damage to membrane lipids (lipid peroxidation; LPO) in porcine thyroid homogenates was induced by Fenton reaction substrates (Fe^2+^ + H_2_O_2_). Iron (Fe^2+^) was used in very high concentrations of 1200, 600, 300, 150, 75, 37.5, 18.75, 9.375, 4.687, and 2.343 µM. Indole-3-butyric acid (10.0, 5.0, 2.5, 1.25, and 0.625 mM) was applied to check whether it prevents the above process. The LPO level, expressed as malondialdehyde + 4-hydroxyalkenals (MDA + 4-HDA) concentration, was measured spectrophotometrically. Expectedly, Fenton reaction substrates, in a Fe^2+^ concentration-dependent manner, increased LPO level, with the lowest effective concentration of iron being 9.375 µM. In the case of almost all concentrations of indole-3-butyric acid, this auxin has exhibited very promising antioxidant protection, with the most effective concentrations being 10.0 and 5.0 mM; however, as low concentrations of indole-3-butyric acid at 1.25 mM was still effective. Indole-3-butyric acid used alone did not change the basal level of LPO, which is a favourable effect. To summarise, indole-3-butyric acid has protective antioxidant properties against experimentally induced oxidative damage to membrane lipids in the thyroid, and this is for the first time documented in the literature. This compound can be considered a natural protective agent present in plants, which can serve as a dietary nutrient.

## 1. Introduction

Antioxidants are compounds that prevent the production of free radicals or neutralise them, and, in this way, they can decrease the oxidative degradation of biomolecules [1]. A lack of balance between antioxidant defence and the production of reactive oxygen species (ROS) can contribute to the development of serious diseases such as cancer, retinopathy, asthma, pulmonary hypertension, etc. ROS are able to interact with proteins, lipids, and DNA [2]. Fenton reaction (Fe^2+^ + H_2_O_2_→ Fe^3+^ + ^•^OH + OH^−^) is a very popular example of oxidative stress reactions in which hydroxyl radical (^•^OH) is formed [3]. This model of reaction is often applied to experimentally induce lipid peroxidation (LPO) [4,5,6]. LPO is a result of membrane phospholipid damage caused by ROS [7]. This process consists of three steps such as initiation, transmission, and termination. The first step of the whole process is the initiation when the hydrogen atom is detached by ^•^OH from the unsaturated fatty acid of phospholipids, and alkyl radicals (L^•^) are formed next. The formation of conjugated bonds is followed by rearrangement of double bonds. In the next step, i.e., transmission, free alkyl radicals may react with oxygen or fatty acids and that leads to the formation of new lipid-free radicals (peroxyl radical, LOO^•^). It is a chain reaction that can be an effect of autooxidation of several hundred molecules of polyunsaturated fatty acids. Termination is the last step when a product without a free radical is formed [8]. Finally, fatty acid dimers and oxo- or hydroxyl-fatty acids with a modified structure are the products of LPO. Further transformations of these products lead to the formation of malondialdehyde (MDA), hydroxyaldehydes, and 4-hydroxy-2-nonenal (4-HNE), which are highly reactive aldehydes. These compounds can react with amino acid residues of enzymes and that results in disruption of their structure and functioning. MDA and 4-HNE can interact with DNA, and that leads to disruptions in DNA replication and transcription [9]. MDA and 4-HNE inhibit the activity of antioxidant enzymes and influence the activity of signalling proteins between cells or dysregulate metabolic processes [10,11,12]. Mitochondria are particularly sensitive to LPO [13]. Moreover, oxidative changes of membrane phospholipids in mitochondrion disrupt the electron transport chain and lead to increased production of ROS [14,15]. It should be added that 4-HNE, similar to 4-hydroxy-2E-hexenal (4-HHE) and 4-hydroxy-2E,6Z-dodecadienal (4-HDDE), belongs to the group of 4-hydroxyalkenal (4-HDA) [16], which is measured in the current study.

Indole-3-butyric acid (IBA) is a chemical compound that naturally occurs in *Arabidopsis thaliana*. It is known that indole-3-butyric acid is an indole derivative with butyric acid at rest. In general, indoles have many favourable properties, such as antitumour, antiviral (anti-HIV, anti-COVID-19), anti-inflammatory, antidiabetic, and antioxidant, which means that indoles can be applied in medicine [17]. Indole-3-propionic acid (IPA) and melatonin are examples of indole derivatives that have great potential in medicine. The former indole substance is a metabolite of tryptophan in the intestinal microflora. The potential effect of indole-3-propionic acid as a factor influencing body mass reduction is interesting because studies on rats whose diets varied in tryptophan content allowed us to confirm this hypothesis [18]. In turn, melatonin (N-acetyl-5-methoxytryptamine) has numerous favourable effects in humans, such as antioxidative effects; it is safe, and it is used as a supplement [19]. Indole-3-butyric acid belongs to auxins, i.e., plant hormones, that initiate root formation and are used in plant tissue culture to modulate general root and shoot architecture, organ patterning, vascular development, and growth in tissue culture and tropic responses to light and gravity. Therefore, indole-3-butyric acid is typically used to induce rooting when propagating plants by striking or cutting [20].

The aim of the research was to check if oxidative damage to membrane lipids known as LPO, induced by Fe^2+^ + H_2_O_2_ (which are Fenton reaction substrates) in thyroid homogenates, can be prevented by the potential antioxidative effects of indole-3-butyric acid. The study was performed on porcine thyroid tissue homogenates. Iron, as one of the Fenton reaction substrates, was used in the highest achievable concentrations, i.e., approximately ten-fold higher than it was routinely used in most published studies. Although indole-3-butyric acid is not present in the thyroid gland, the thyroid tissue was chosen because this endocrine gland seems to stand out most among all endocrine glands concerning oxidative processes, as suggested by the results of our earlier studies [21].

## 2. Materials and Methods

### 2.1. Ethical Matters

The Polish Act on the Protection of Animals Used for Scientific or Educational Purposes was published on 15 January 2015. It implements Directive 2010/63/EU of the European Parliament and the Council (published on 22 September 2010) on the protection of animals used for scientific purposes. In agreement with the above documents, it is not required that the collection of animal tissues and/or organs should be approved by the local ethics committee. Such animals are registered by the centre in which the collection of tissues/organs is performed. In the present study, selected tissues were taken at a slaughterhouse during the process of slaughter, which is routinely performed for consumption purposes. It should be stressed that we did not use experimental animals in the current study.

### 2.2. Chemicals

Indole-3-butyric acid, hexahydrated iron sulfate (FeSO_4_·6H_2_O), and hydrogen peroxide (H_2_O_2_) were obtained from Sigma (St. Louis, MO, USA), and ethanol (96%) was obtained from Stanlab (Lublin, Poland). An LPO-586 kit for investigation of LPO was purchased from Enzo Life Science (Farmingdale, NY, USA). All chemicals came from commercial sources, and they were of analytical grade. The final concentrations of indole-3-butyric acid in the samples were 0.0 mM, 10.0 mM, 5.0 mM, 2.5 mM, 1.25 mM, and 0.625 mM.

### 2.3. Animals and Tissue Preparation

Porcine thyroids were collected at a slaughterhouse. To protect animals at the time of killing, all the requirements of the European Community Council Regulation (CE1099/2009) were met. The age of the animals was approx. 8–9 months; therefore, they were sexually mature. Their body mass was 120.15 ± 5.2 (SD) kg. According to the statement of the veterinary medical officer, who is responsible for the health of the animals and the hygiene of the slaughterhouse, all the animals were free of pathologies and, generally, they were in good body condition. The thyroid tissue was collected immediately after the slaughter and frozen on solid CO_2_. Then, it was stored at −80 °C until the experimental procedure.

### 2.4. Preparation of Samples with Porcine Thyroid Homogenates

An ice-cold 50 mM Tris-HCl buffer (pH 7.4) (10%, *w*/*v*) was applied to homogenise thyroid tissues. Then, homogenates were incubated at 37 °C (30 min) in the presence of selected substances. Indole-3-butyric acid was dissolved in absolute ethanol, and the obtained concentrations were 0.0 mM, 10.0 mM, 5.0 mM, 2.5 mM, 1.25 mM, and 0.625 mM. The final concentration of ethanol during incubation was 1% (*v*/*v*). The concentrations of FeSO_4_·6H_2_O in the final experiments were 0.0, 1200, 600, 300, 150, 75, 37.5, 18.75, 9.375, 4.687, and 2.343 µM. It should be mentioned that in the pilot experiments, we used FeSO_4_ in as high a concentration as 2400 µM; however, due to interaction with the method, the concentration of 2400 µM was not taken into consideration in the final analyses. The concentration of hydrogen peroxide (H_2_O_2_) used was 5 mM. The above-described experiment was performed three times.

### 2.5. Lipid Peroxidation Assay

The concentrations of malondialdehyde + 4-hydroxyalkenals (MDA + 4-HDA) did constitute the index of LPO. An LPO-586 kit (Enzo Life Science, Farmingdale, NY, USA) was applied to measure the level of MDA + 4-HDA. After incubation of the homogenates with examined substances, centrifugation was performed (5000× *g*, 10 min, 4 °C) to obtain supernatant. Thereafter, the experiment with the use of supernatant was performed in duplicate in the following way. The supernatant (200 μL) was mixed with methanol:acetonitrile (1:3, *v*/*v*) solution (650 μL), which contained N-methyl-2-phenylindole being a chromogenic reagent. Then, the mixture was vortexed. In the next step, methanesulfonic acid (15.4 M, 150 μL) was added to the mixture, and thereafter, the incubation was performed (45 °C, 40 min). As a result of the reaction between MDA + 4-HDA and N-methyl-2-phenylindole, a chromophore was produced. It was spectrophotometrically measured at an absorbance of 586 nm, using a solution of 10.0 mM 4-hydroxynonenal as the standard. The amount of MDA + 4-HDA (nmol) per mg of protein expressed the LPO level. The amount of protein was determined by the use of Bradford’s method, with the standard being bovine albumin.

### 2.6. Statistical Calculations

Statistical analyses were performed with the application of SigmaPlot 11.0. The following statistical tests were used: a one-way analysis of variance (ANOVA), followed by the Student–Neuman–Keuls’ test, and an unpaired *t*-test. The level of *p* < 0.05 determined statistical significance. The obtained results are shown as means ± SE.

## 3. Results

Herein are presented results on LPO in porcine thyroid homogenates induced by FeSO_4_·6H_2_O in ten different concentrations plus hydrogen peroxide (5.0 mM) for induction of hydroxyl radicals. As a potential protective substance, an indole derivative was applied, which was indole-3-butyric acid used in six different concentrations. Figure 1 presents the concentration of malondialdehyde and 4-hydroxyalkenals (MDA + 4-HDA) in porcine thyroid homogenates incubated with FeSO_4_·6H_2_O (0.0, 1200, 600, 300, 150, 75, 37.5, 18.75, 9.375, 4.687, and 2.343 µM) with the addition of H_2_O_2_ (5.0 mM). As expected, Fenton reaction substrates induced LPO in a concentration-dependent manner, with a statistical significance observed for FeSO_4_·6H_2_O in concentrations of 1200-9.375 µM.

As it is presented in Figure 2, indole-3-butyric acid is an effective antioxidant agent in the four highest concentrations, i.e., 10.0 mM, 5.0 mM, 2.5 mM, and 1.25 mM, against iron concentrations of 1200 µM, 18.74 µM, and 9.375 µM. Regarding other iron concentrations that induced LPO, it was reduced by either two or three of the highest concentrations of indole-3-butyric acid.

A cumulative graph, shown in Figure 3, constitutes the summary of the protective effect of the tested indole compound used in different concentrations on the level of LPO in individual samples containing different concentrations of iron ions.

Figure 4 presents that indole-3-butyric acid, added alone to the incubation medium, did not cause statistically significant changes regarding the level of LPO, and this suggests that this indole substance did not cause additional interactions in samples with the homogenate.

## 4. Discussion

Oxidative stress parameters are examined in response to numerous nutritional factors in both human and experimental studies [22,23].

In this paper, we described the results of our investigation of the antioxidative properties of indole-3-butyric acid in porcine thyroid homogenates. As it is known, indole-3-butyric acid is generally used in experimental studies as an auxin and many scientific publications concern this issue. Here, we decided to focus on an examination of the potential medicinal properties of this compound, namely its protective antioxidant activity. For this reason, we have used an LPO assay and, as a prooxidative factor, we have chosen FeSO_4_·6H_2_O and H_2_O_2_ to initiate the Fenton reaction in porcine thyroid homogenates; we have chosen ten different concentrations of FeSO_4_·6H_2_O (0.0, 1200, 600, 300, 150, 75, 37.5, 18.75, 9.375, 4.687, 2.343 µM) and used H_2_O_2_ in the concentration of 5.0 mM. Choosing iron together with the thyroid gland was justified because iron is an indispensable element for thyroid hormone synthesis. A relationship between iron deficiency and thyroid function has been broadly explored and defined [24]. However, when in excess, this micronutrient can be deleterious, causing oxidative damage.

Physiological iron concentration in human blood serum is approximately 11–29 μmol/L in women and 14–32 μmol/L in men [25,26,27]. Thus, concentrations used in the present study, at least those of 37.5 µM and lower, correspond to physiological conditions in humans. In turn, higher concentrations of iron used in the present study, i.e., 75–1200 µM, correspond to conditions associated with hemochromatosis [27].

As mentioned before, we have examined indole-3-butyric acid as a potential free radical scavenger, and we have used it in five different concentrations of 10.0, 5.0, 2.5, 1.25, and 0.625 mM. There are few data published in the literature regarding the antioxidative effects of indole-3-butyric acid. Topalca et al. [28] examined the antioxidant effect of indole-3-butyric acid in rats when this auxin was used in two doses, i.e., 25 and 50 ppm. Activities of known antioxidant enzymes such as superoxide dismutase (SOD), glutathione peroxidase (GSH-Px), and glutathione-S-transferase (GST), were evaluated in different tissues (brain, erythrocytes, kidney, heart, liver, lungs, muscle, spleen) and it was documented that indole-3-butyric acid increased levels of SOD and GST in the examined tissues. On the opposite end, another investigation by Yilmaz et al. [29] revealed that indole-3-butyric acid had an immunotoxic effect in rats because it inhibited adenosine deaminase and increased the level of myeloperoxidase in rat tissues. Moreover, the activities of acetylcholinesterase and butyrylcholinesterase were diminished in some rat tissues, and this is the reason why indole-3-butyric acid may be suspected of causing neurotoxic effects [29]. In another study, Pattayil et al. [30] investigated apoptotic properties of butyric acid derivatives, including indole-3-butyric acid, in colorectal carcinoma cells (HCT116). It has been shown that indole-3-butyric acid is cytotoxic in these cancer cells almost at a similar level as a known cytostatic—5-fluorouracil, and this is obviously a favourable effect [30].

It is also worth mentioning that a health and environmental risk assessment, according to the U.S. Environmental Protection Agency, revealed that indole-3-butyric acid may have a harmful effect on human organs, such as eyes and skin, and may cause inhalation irritation. At the same time, it is important to emphasise that this agency recommends applying indole-3-butyric acid in tiny amounts [31]. Accordingly, as indicated by the results of our research, indole-3-butyric acid showed very promising antioxidant activity.

Several similar studies were performed before with the use of other indole substances, such as melatonin and indole-3-propionic acid; they were applied as potential antioxidant agents, and the highest used concentrations were 5.0 and 7.5 mM, respectively. Melatonin and indole-3-propionic acid are also indole derivatives and exhibited protective effects against experimentally induced LPO even when iron was used at high concentrations at 1200 µM [4,5]. Because indole-3-butyric acid effectively reduced oxidative damage to membrane lipids induced by high iron concentrations, this auxin can be considered a promising antioxidant agent in further research.

The concentrations of the indole-3-butyric acid used in the present study were similar to concentrations of melatonin and indole-3-propionic acid used in previous studies [4,5]. These concentrations of indole-3-butyric acid should be referred to its levels found in the compartments of living organisms or to recommended safe doses. Regarding the first issue physiological levels of indole-3-butyric acid are not known. According to the Environmental Protection Agency, this compound exhibits very good activity at very low concentrations, which is often several orders of magnitude below 1% (equal to 0.0615 M), i.e., the concentration that seems to be safe [31]. When this compound was tested with the use of HCT116 cancer cells, it was shown that the half maximal inhibitory concentration values (IC_50_) were in the range of 6.28–4.39 mM [30], and these concentrations are comparable to those used in the present study. Such a comparison allows us to assume that indole-3-butyric acid in doses similar to those applied in the present study can be effective in cancerous cells. However, this hypothesis should be proved in future studies.

Mechanisms of protective effects of indole-3-butyric acid should be considered. Regarding the potential effects of indole-3-butyric acid on oxidative reactions, this auxin could neutralise free radicals and reactive oxygen species but also affect antioxidative and prooxidative enzymes. However, data on these issues in the literature are limited. Unfortunately, there are no published results on the direct effects of indole-3-butyric acid on reactive oxygen species and on free radicals, especially on hydroxyl radicals, because this free radical was generated by the Fenton reaction substrate in the current study. We suspect that the most likely mechanism of action of indole-3-butyric acid is the activation of enzymes important in the cell defence against oxidative stress. As it was mentioned before, indole-3-butyric acid-activated antioxidant defence systems in rat tissues, especially GST and SOD, were seriously influenced by indole-3-butyric acid because a significant increase in their activities was observed in almost all analysed tissues [28].

It should also be discussed how indole-3-butyric acid effected the basal level of LPO. Of great importance is the fact that this auxin added to thyroid homogenates did not significantly change LPO. We have observed the same in most in vitro studies with the use of melatonin [4,5]. It is reasonable to state that an excellent antioxidant does not change the level of oxidative processes (the oxidative damage to macromolecules included) under physiological conditions. Regarding melatonin, this indoleamine was proven not to change the basal level of oxidative damage in in vivo studies [32], but regarding indole-3-butyric acid, such an effect should be checked in future in vivo experiments. More detailed and interesting studies can be planned for subsequent experiments, which may allow for the assessment of the effects of indole-3-butyric acid not only on other oxidative processes but also on parameters such as calcium levels and ATP.

Our study has, of course, some limitations. The first limitation relates to the fact that our model was an in vitro model with the use of tissue homogenates. Therefore, it should be stated that the results obtained in the present study can not be directly extrapolated to the in vivo conditions, especially to human organisms. The next limitation results from the application of only one kind of tissue, i.e., the thyroid gland. Because this endocrine gland is characterised by a high level of oxidative stress [21], it is not excluded that results obtained in other endocrine glands or any other organs would be different. It should also be stated that examining protective effects was limited to three selected antioxidants, i.e., two indole substances and 17β-estradiol.

Regarding the practical application of indole-3-butyric acid, the greatest opportunities are in the area of agriculture, all types of plants such as vegetables and fruits breeding and cultivation. Indole-3-butyric acid, as a plant growth regulator, induces differentiation and growth of root tissue and improves absorption of nutrient elements. The experiment of Khadr et al. showed that indole-3-butyric acid at concentrations of 100 and 150 µM caused significant growth of carrot roots [33]. Another interesting study has shown that indole-3-butyric acid at a concentration equal to 10^−11^ M effectively stimulated the growth of maize plants [34]. In the same study, the use of indole-3-butyric acid at a concentration of 10^−9^ M reversed the unfavourable effects of cadmium ions, which caused a decrease in nutrients and an increase of H_2_O_2_ concentration [34]. Another practical aspect relates to the fact that LPO, the process evaluated in the current study, is one of the pathways leading to rancidity in food. Therefore, indole-3-butyric acid can act as an antioxidant at two levels, i.e., before food consumption and in the living organism.

Also, potential favourable effects on physiological systems should be mentioned. Taking into account the results of the present study, it can be supposed that exposure of a living organism to indole-3-butyric acid may contribute to keeping the balance between ROS production and defence against them at cellular, tissue and, possibly, system levels, which consequently protects against various and serious diseases.

To sum up, there is a thin border between the toxicity and beneficial effects of this auxin; therefore, it should be kept when using it, especially because it can reach the bodies of humans and animals indirectly. This compound can be considered as a natural protective agent present in plants, as an additive in the food industry, and also as a supplement recommended to individuals suffering from disorders which result from iron excess or—generally—from increased oxidative stress.

According to our knowledge, the current study is the first one to document the antioxidant effects of indole-3-butyric acid against experimentally induced oxidative damage in animal tissues.

## 5. Conclusions

Indole-3-butyric acid has protective antioxidant properties against membrane lipids in the thyroid and this is for the first time documented in the literature. No prooxidative effects of this auxin suggest that it is safe for mammals in terms of affecting oxidative processes. This compound can be considered a natural protective agent present in plants, which can serve as a dietary nutrient.

## Figures and Tables

**Figure 1 nutrients-16-03010-f001:**
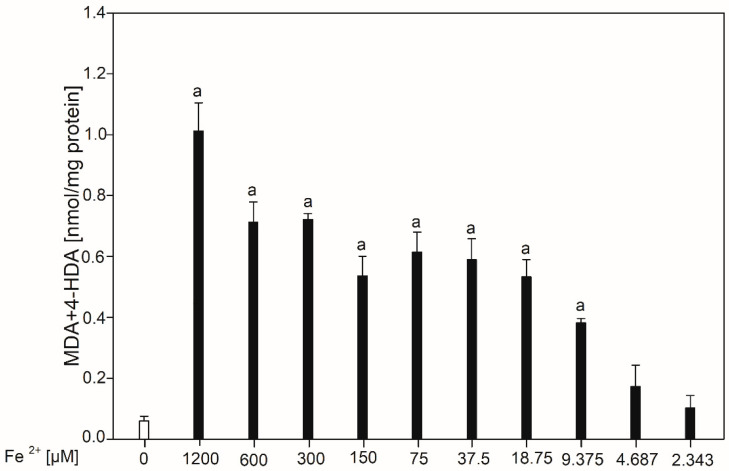
Concentration of malondialdehyde and 4-hydroxyalkenals (MDA + 4-HDA) in porcine thyroid homogenates. Homogenates were incubated in the presence of FeSO_4_·6H_2_O (0.0, 1200, 600, 300, 150, 75, 37.5, 18.75, 9.375, 4.687, 2.343 µM) plus H_2_O_2_ (5.0 mM) used to induce LPO. Data are expressed as the amount of MDA + 4-HDA (nmol) per mg protein. Bars represent the mean ± SE of three independent experiments run in duplicates. ^a^ *p* < 0.05 vs. control ‘0’.

**Figure 2 nutrients-16-03010-f002:**
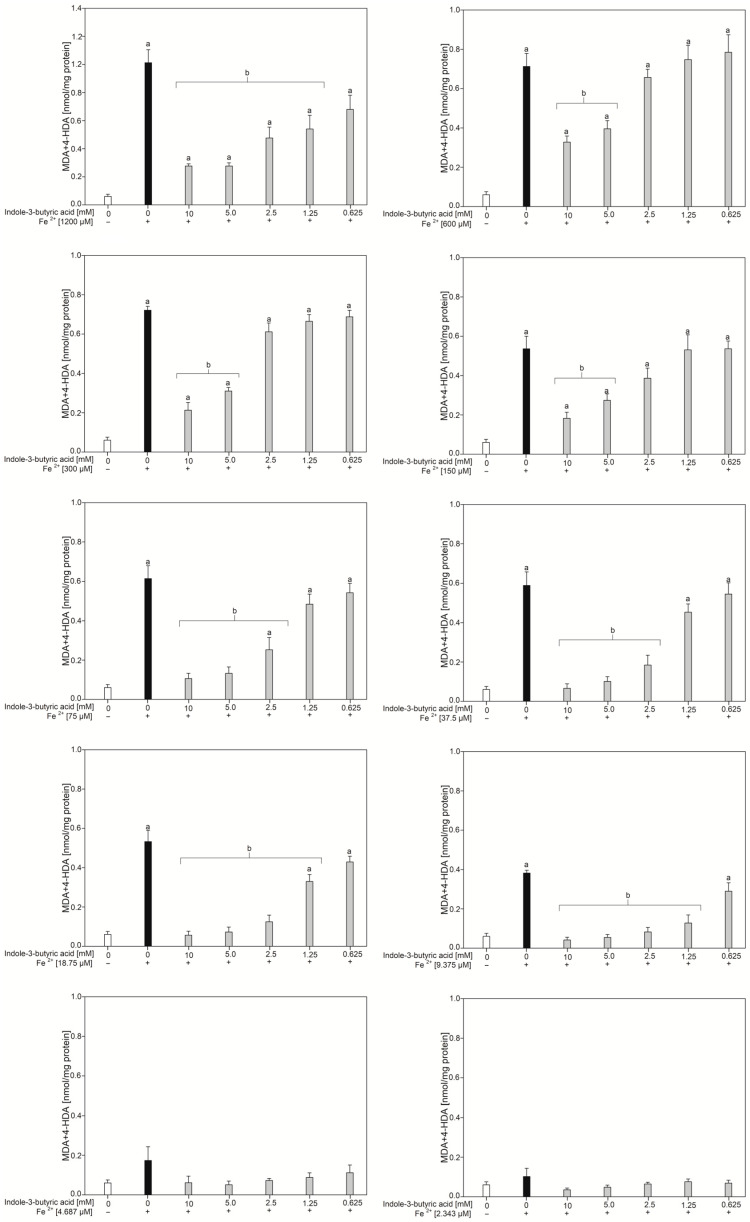
Concentration of malondialdehyde + 4-hydroxyalkenals (MDA + 4-HDA) in the porcine thyroid homogenates, which were incubated in the presence of different concentrations of FeSO_4_·6H_2_O (0.0, 1200, 600, 300, 150, 75, 37.5, 18.75, 9.375, 4.687, 2.343 µM) plus H_2_O_2_ (5.0 mM) and, additionally, in the presence of indole-3-butyric acid (0.0, 10.0, 5.0, 2.5, 1.25, 0.625 mM). Data are expressed as the amount of MDA + 4-HDA (nmol) per mg of protein. Bars represent the mean ± SE of three independent experiments run in duplicate. ^a^
*p* < 0.05 vs. control ‘0’; ^b^
*p* < 0.05 vs. respective concentration of Fe^2+^ ions with addition of H_2_O_2_ (without indole-3-butyric acid).

**Figure 3 nutrients-16-03010-f003:**
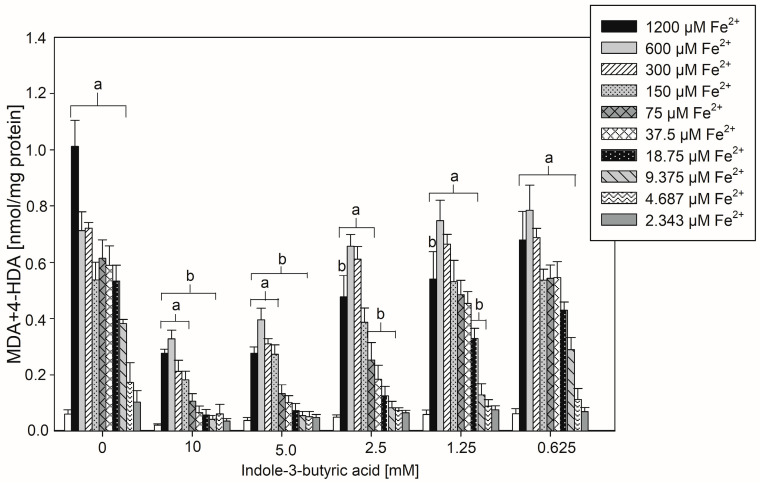
The concentrations of malondialdehyde + 4-hydroxyalkenals (MDA + 4-HDA) in porcine thyroid homogenates. Homogenates were incubated in the presence of FeSO_4_·6H_2_O and H_2_O_2_ used to induce LPO and, additionally, in the presence of indole-3-butyric acid (0.0, 10.0, 5.0, 2.5, 1.25, 0.625 mM). Data are expressed as the amount of MDA + 4-HDA (nmol) per mg of protein. Bars represent the mean ± SE of three independent experiments run in duplicates. ^a^ *p* < 0.05 vs. control ‘0’; ^b^ *p* < 0.05 vs. respective concentration of Fe^2+^ plus H_2_O_2_ (without indole-3-butyric acid).

**Figure 4 nutrients-16-03010-f004:**
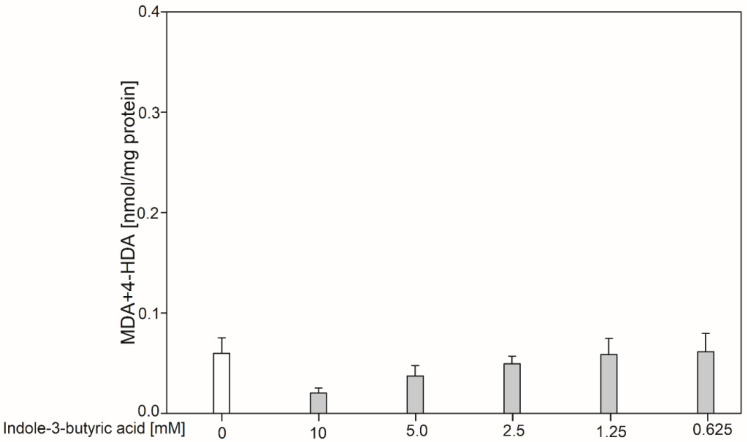
Concentration of malondialdehyde and 4-hydroxyalkenals (MDA + 4-HDA) in porcine thyroid homogenates. Homogenates were incubated in the presence of indole-3-butyric acid (0.0, 10.0, 5.0, 2.5, 1.25, 0.625 mM). Data are expressed as the amount of MDA + 4-HDA (nmol) per mg protein. Bars represent the mean ± SE of three independent experiments run in duplicates.

## Data Availability

The raw data supporting the conclusions of this article will be made available by the authors on request due to privacy.

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
