# Peer review of "Indole-3-Butyric Acid, a Natural Auxin, Protects against Fenton Reaction-Induced Oxidative Damage in Porcine Thyroid"

_nutrients, 2024, doi:10.3390/nu16173010_

Round 1
Reviewer 1 Report
Comments and Suggestions for Authors
Is the fIg3 a combined figure of all figures in Fig2? If so, please use either of them?
Does the author think that the antioxidant activity of the Indole 3 butyric acid is due to the scavenging of radical by indole ring? If so, any other indole derivatives would also have a similar effect. Please use another indole derivative to test this.
The authors should mention how results could benefit physiological systems.
Is Indole 3 butyric acid present in thyroid gland? Mention why was it chosen for thyroid gland?
The authors used several concentrations of the indole 3 butyric acid. They must comment on the optimum concentration of the compound that was efficient in controlling the free radical generation and how that could be relevant in human body.
The conclusion should be shortened and be relevant to the results obtained.
Author Response
Dear Editors,
Thank you very much for giving us the opportunity to resubmit the paper entitled “Indole-3-butyric acid, a natural auxin, protects against Fenton reaction-induced oxidative damage in porcine thyroid”. According to the Reviewers remarks, which we found to be very helpful, we have introduced changes to the first version of the manuscript. The changes are specified below.
Reviewer 1
- Is the fIg3 a combined figure of all figures in Fig2? If so, please use either of them?
We would like to thank very much the Reviewer for this comment. That was a point of our consideration whether we should present the results in two different ways. Finally, we decided to prepare this cumulative graph, as it helps to understand rather complex scheme of the study design. At the same time the results showing in Fig 2, specifying effects of indole-3-butyric acid against particular iron concentrations, are of value because they give better possibility to easier identify statistically significant differences. Therefore, we stated: “A cumulative graph, shown in the Figure 3, constitutes the summary of the protective effect of the tested indole compound, used in different concentrations, on the level of LPO in individual samples containing different concentrations of iron ions.” (line 164-166)
At this moment we decided not to remove any of these two figures. However, if the Reviewer still requires we will do this in the next revision.
- Does the author think that the antioxidant activity of the Indole 3 butyric acid is due to the scavenging of radical by indole ring? If so, any other indole derivatives would also have a similar effect. Please use another indole derivative to test this.
We would like to thank very much the Reviewer for this very important remark. Because many, if not most, indole substances reveal the ability to scavenge free radicals, it is highly probable that this phenomenon is due to the presence of indole ring. However, this issue is not clearly documented in the literature. Regarding other indole derivatives, such as for example melatonin and indole-3-propionic acid, they are well known free radical scavengers. Because indole-3-butyric acid, which was examined in the present study, is an auxin, also other indole derivatives produced by plants should be considered. However, this area has not been explored till now. It should be stressed that the mechanisms of antioxidative effects of indole-3-butyric acid is till now very poorly recognized, which is discussed by us in the Discussion section (lines 257-268).
As the Reviewer suggested, we plan to perform further studies with the use of such indole derivatives as, for example, indole-3-acetic acid, which also belongs to plant growth factors (auxins), and publish the results of these studies in a separate paper.
- The authors should mention how results could benefit physiological systems.
We would like to thank very much the Reviewer for this comment. Many diseases result from oxidative stress. It is important to thoroughly investigate and analyze the mechanisms of action of chemical compounds that have an inhibitory effect on the formation and activity of reactive oxygen species. Indoles are a group of compounds that are the object of our interest. They can be potentially used in medicine.
According to the Reviewer’s suggestion a short paragraph has been added in the Discussion section regarding this issue (lines 300-305): “Also potential favorable effects on physiological systems should be mentioned. Taking into account results of the present study it can be supposed that exposure of a living organism to indole-3-butyric acid may contribute to keeping the balance between ROS production and defense against them at cellular, tissue and, possibly, system levels, which consequently protects against various and serious diseases.”.
- Is Indole 3 butyric acid present in thyroid gland? Mention why was it chosen for thyroid gland?
We would thank very much the Reviewer for this comment. Indole-3-butyric acid is not present in thyroid gland and this information we has been added in the Introduction section (lines 81-84): ”Although indole-3-butyric acid is not present in thyroid gland, the thyroid tissue was chosen because this endocrine gland seems to most stand out among all endocrine glands concerning oxidative processes, as suggested by the results of our earlier studies [23].”
- The authors used several concentrations of the indole 3 butyric acid. They must comment on the optimum concentration of the compound that was efficient in controlling the free radical generation and how that could be relevant in human body.
We would like to thank very much the Reviewer for this comment. It was not easy for us to choose the optimal concentration of indole-3-butyric acid, because we had not previously conducted experiments with this substance. For this reason, a number of concentrations of this auxin were used by us in pilot studies to choose the optimal range of concentrations for the final study. At the same time it should be stressed that the range of concentrations of indole-3-butyric acid, applied of the current study, is similar to those used in case of melatonin or indole-3-propionic acid.
The answer to the Reviewer’s question is present in the following paragraph (lines 244-256): “The concentrations of the indole-3-butyric acid used in the present study were similar to concentrations of melatonin and indole-3-propionic acid used in previous studies [5,6]. These concentrations of indole-3-butyric acid should be referred to its levels found in the compartments of living organisms or to recommended safe doses. Regarding the first is-sue physiological levels of indole-3-butyric acid are not known. According to Environ-mental Protection Agency this compound exhibits very good activity at very low concentrations, which is often several orders of magnitude below 1% (equal to 0,0615 M), i.e. the concentration which seems to be safe [33]. When this compound was tested with the use of HCT116 cancer cells, it was shown that the half maximal inhibitory concentration values (IC50) were in the range of 6.28-4.39 mM [32] and these concentrations are comparable to those used in the present study. Such a comparison allows to assume that in-dole-3-butyric acid in doses similar to those applied in the present study can be effective in cancerous cells. However, this hypothesis should be proved in the future studies.”
- The conclusion should be shortened and be relevant to the results obtained.
We absolutely agree with the Reviewer’s remark. Therefore, we have shortened the conclusion by moving its part to the Discussion section (lines 307-310). The conclusion has also been shortened in the Abstract.
Reviewer 2 Report
Comments and Suggestions for Authors
In the manuscript, the authors has revealed that indole -3-butyric acid, a naturally occurring auxin, guards against oxidative damage brought on by the Fenton reaction, in pig thyroid.
Although the manuscript is interesting it needs improvement.
1. Authors should consider improving their abstract writing as it is unclear.
2. The authors should revise the introduction. When they first mention “lipid peroxidation (LPO)” (l. 39) they should then consider including only the abbreviation when citing it again (i.e. l ​​40). There are also several editing errors and English errors.
3. In the introduction, the authors mention 4-HDA, stating that they measure it in the aforementioned article. Here, the authors should describe in detail the compound used (l. 60).
4. In materials and methods, the authors state that they did not perform the removal of porcine thyroid tissue. However, they should describe the procedure in which it occurred (is there a standardized process?) and define the status of the animal at the time of killing, or at least some more detail. In general, they should re-evaluate the editing of paragraphs included in materials and methods.
5. Regarding statistics, what program did they use?
Although the work is interesting it should be supported by other experiments such as evaluating the levels of calcium or ATP produced.
Author Response
Dear Editors,
Thank you very much for giving us the opportunity to resubmit the paper entitled “Indole-3-butyric acid, a natural auxin, protects against Fenton reaction-induced oxidative damage in porcine thyroid”. According to the Reviewers remarks, which we found to be very helpful, we have introduced changes to the first version of the manuscript. The changes are specified below.
In the manuscript, the authors has revealed that indole-3-butyric acid, a naturally occurring auxin, guards against oxidative damage brought on by the Fenton reaction, in pig thyroid.
Although the manuscript is interesting it needs improvement.
- Authors should consider improving their abstract writing as it is unclear.
We would like to thank very much the Reviewer for this suggestion. We have made appropriate changes in the Abstract.
- The authors should revise the introduction. When they first mention “lipid peroxidation (LPO)” (l. 39) they should then consider including only the abbreviation when citing it again (i.e. l ​​40). There are also several editing errors and English errors.
We absolutely agree with the Reviewer’s remark. We have used the abbreviation “LPO” in the Introduction section to avoid repeating the phrase “lipid peroxidation” in lines: 40; 50; 57 and 78. We have also corrected other errors.
- In the introduction, the authors mention 4-HDA, stating that they measure it in the aforementioned article. Here, the authors should describe in detail the compound used (l. 60).
We would like to thank very much the Reviewer for this remark. In the current study, to evaluate the level of lipid peroxidation, we have used LPO-586 kit. This kit measures malondialdehyde+4-hydroxyalkenals (MDA+4-HDA). 4-HDA is a group of lipid peroxidation products, such as 4-hydroxy-2E-hexenal (4-HHE), 4-hydroxy-2E-nonenal (4-HNE) and 4-hydroxy-2E,6Z-dodecadienal (4-HDDE) (line 57). However, it should be underlined that the biochemical kit used in the current study does not identify a specific kind of 4-HDA.
It should be stressed that our study focuses on protective antioxidative effects of indole-3-butyric acid. Therefore, we hope that the short information on 4-HDA, added to the text, will be satisfactory for the Reviewer. However, if the Reviewer still requires we can provide an accurate description of this issue in the next revision.
- In materials and methods, the authors state that they did not perform the removal of porcine thyroid tissue. However, they should describe the procedure in which it occurred (is there a standardized process?) and define the status of the animal at the time of killing, or at least some more detail. In general, they should re-evaluate the editing of paragraphs included in materials and methods.
We absolutely agree with the Reviewer’s suggestion. Therefore, we have added the following paragraph (lines 107-115):
“Porcine thyroids were collected from animals at a slaughterhouse. Animals were treated according to the European Community Council Regulation (CE1099/2009) concerning the protection of animals at the time of killing. All animals were sexually mature as determined by age (8–9 months) and body mass [120.15 ± 5.2 (SD) kg]. They were in good body condition and considered free of pathologies by the veterinary medical officer responsible for the health of the animals and the hygiene of the slaughterhouse. Immediately (in less than 5 min) after the slaughter, the thyroid tissue was collected, frozen on solid CO2, and stored at −80°C till experimental procedure.”
- Regarding statistics, what program did they use?
We would thank very much the Reviewer for this remark. We used Sigma Plot 11.0 for statistical calculations and graphs preparations. The statement is as follows (lines 139-141): “The data were statistically analysed with the application of Sigma Plot 11.0, using a one-way analysis of variance (ANOVA), followed by the Student–Neuman–Keuls’ test, or using an unpaired t-test. Statistical significance was determined at the level of p < 0.05. Results are presented as means ± SE.”
Although the work is interesting it should be supported by other experiments such as evaluating the levels of calcium or ATP produced.
We absolutely agree with the Reviewer’s suggestion. Such more detailed and interesting studies can be planned for performance in subsequent experiments, thanks to which it will be possible to assess the effect of indole-3-butyric acid on such parameters, as the levels of calcium or ATP produced.
It should be stressed that the current study is the first one on antioxidative protective effects of indole-3-butyric acid. With the scientific basis presented in our original paper it will be easier and justified to arrange future studies, in which calcium and ATP levels will be examined together with parameters of oxidative stress.
Round 2
Reviewer 1 Report
Comments and Suggestions for Authors
The paper can be accepted in its current form.
Author Response
Dear Reviewer,
Thank you for your positive feedback and recommendation for acceptance. We appreciate the time and effort you have taken to review our manuscript.
Sincerely
Małgorzata Karbownik-Lewińska